# Evaluation of Feature Selection Techniques for Breast Cancer Risk Prediction

**DOI:** 10.3390/ijerph182010670

**Published:** 2021-10-12

**Authors:** Nahúm Cueto López, María Teresa García-Ordás, Facundo Vitelli-Storelli, Pablo Fernández-Navarro, Camilo Palazuelos, Rocío Alaiz-Rodríguez

**Affiliations:** 1Department of Electrical, Systems and Automatic Engineering, Universidad of León, Campus de Vegazana s/n, 24071 León, Spain; ncuetl00@estudiantes.unileon.es (N.C.L.); rocio.alaiz@unileon.es (R.A.-R.); 2Centro de Investigación Biomédica en Red (CIBER), Grupo Investigación Interacciones Gen-Ambiente y Salud (GIIGAS), Instituto de Biomedicina (IBIOMED), Universidad de León, 24071 León, Spain; facuvitelli17@gmail.com; 3Cancer and Environmental Epidemiology Unit, National Center for Epidemiology, Carlos III Institute of Health, 28903 Madrid, Spain; pfernandezn@isciii.es; 4Consortium for Biomedical Research in Epidemiology and Public Health (CIBERESP), 28029 Madrid, Spain; 5Department of Mathematics, Statistics, and Computing, University of Cantabria-IDIVAL, 39005 Santander, Spain; camilo.palazuelos@alumnos.unican.es

**Keywords:** breast cancer, risk prediction model, feature selection, stability

## Abstract

This study evaluates several feature ranking techniques together with some classifiers based on machine learning to identify relevant factors regarding the probability of contracting breast cancer and improve the performance of risk prediction models for breast cancer in a healthy population. The dataset with 919 cases and 946 controls comes from the MCC-Spain study and includes only environmental and genetic features. Breast cancer is a major public health problem. Our aim is to analyze which factors in the cancer risk prediction model are the most important for breast cancer prediction. Likewise, quantifying the stability of feature selection methods becomes essential before trying to gain insight into the data. This paper assesses several feature selection algorithms in terms of performance for a set of predictive models. Furthermore, their robustness is quantified to analyze both the similarity between the feature selection rankings and their own stability. The ranking provided by the SVM-RFE approach leads to the best performance in terms of the area under the ROC curve (AUC) metric. Top-47 ranked features obtained with this approach fed to the Logistic Regression classifier achieve an AUC = 0.616. This means an improvement of 5.8% in comparison with the full feature set. Furthermore, the SVM-RFE ranking technique turned out to be highly stable (as well as Random Forest), whereas relief and the wrapper approaches are quite unstable. This study demonstrates that the stability and performance of the model should be studied together as Random Forest and SVM-RFE turned out to be the most stable algorithms, but in terms of model performance SVM-RFE outperforms Random Forest.

## 1. Introduction

Nowadays, the occurrence of cancer is steadily increasing [1]. Breast cancer (BC) is the second highest prevalent cancer globally after lung cancer, with 2.09 million cases during 2018 [2]. There are many factors causing this increase; for example, the growth and aging of the population and some other risk factors such as smoking, overweight, physical inactivity, oral contraceptives, economic development, etc. [3,4,5]. Lung cancer is the first cause of cancer death all over the world, but BC remains the leading cause of cancer death among females in less developed countries [6].

The problem of BC has also been addressed in the machine learning filed from the perspective of diagnosis. Thus, there are many studies that focus on extracting meaningful features from different types of data (digital mammography, ultrasound, biopsy) in order to determine whether or not a person has BC. For example, Rajaguru et al. [7] developed a non-invasive method to detect the BC at an early stage using a Gaussian Mixture Model (GMM), obtaining an accuracy of 89.60%, and Radial Basis Function (RBF), obtaining results up to 92.75% of accuracy. In [8], different machine learning algorithms were used: Support Vector Machine (SVM), Decision Tree (C4.5), Naive Bayes (NB) and k-Nearest Neighbors (k-NN) on the Wisconsin Breast Cancer (original) dataset, being the SVM that offers the best results (97.13%). This dataset, the Wisconsin Breast Cancer dataset [9], is composed of features computed from a digitized image of a fine needle aspirate (FNA) of a breast mass. Characteristics of the cell nuclei present in the image such as radius, texture, perimeter, area, etc., are described, and it is used to determine if a tumor is benign or malignant. This analysis implies that the tumor has to be previously detected to determine whether it is malignant or benign, and furthermore, data collection is an invasive technique, as you need to remove a tissue sample. In our work, we try to detect breast cancer risk before the tumor appears.

A similar procedure has been followed in [10], where a Bayesian network (BN) modeling approach has been implemented. The results are promising, but we found the same problem, the method to obtain the data set is an invasive method.

To overcome the problems found by feature-based methods, convolutional neural networks have been lately used for the purpose of diagnosis or evolution. For example, in [11], Wang et al. proposed the use of convolutional neural networks achieving remarkable performance in classification accuracy with results up to 98%. In the same way, Khan et al. proposed a framework for the detection and classification of malignant cells in breast cytology images [12].

More recently, in [13], a deep convolutional neural network is proposed to classify a mammogram as normal or abnormal. Deep learning is also used in [14]. In this case, the authors propose a method capable of detecting breast cancers at a very early stage using computer vision, image processing, medical diagnosis and neural language processing. In [15], the authors aimed to evaluate the performance of a deep learning algorithm to detect breast cancers on chest computed tomography and to validate the results in the internal and external datasets. Bai et al. [16] brings to light a review collecting the ways in which deep learning can be best integrated into breast cancer screening workflows using digital breast tomosynthesis (DBT). The gap in this case is that almost all the methods that exist evaluate samples obtained from different techniques, but always in an intrusive way. Conversely, they evaluate mammogram results after they are done. In our case, we seek to predict the risk of developing breast cancer from easily known variables of each person, which is why it is a non-intrusive technique and is also very important, since we could be detecting a tumor in early stages when the chances of curing it are still great.

In the case of predicting the evolution of the disease, convolutional Neural Networks are also used in axillary lymph node status, which is an important factor for breast cancer staging and treatment planning [17].

Every year, more than million women are diagnosed with BC, and more than half of them will die because of inaccuracies and delays in diagnosis of the disease [18]. That is why risk prediction is so important. The objective of many investigations has been to find factors that affect the development of BC, that is, risk factors.

Morch et al. [19] carried out an experiment in which 1.8 million women were followed on average for 10.9 years. Their results demonstrated that the relative risk of BC among all current users of hormonal contraception was 1.20. This risk increased from 1.09 with less than 1 year of use to 1.38 with more than 10 years of use. After stopping the taking of hormonal contraceptives, the risk of BC still remains higher in women who had used hormonal contraceptives for 5 years or more than in women who had not used hormonal contraceptives. In [20], it is also shown that current use of oral contraceptives carries an excess risk of BC. On the other hand, Graafland et al. [21] found that although there appears to be some risk to develop BC, the absolute risk is small because recent research demonstrates that contraceptives may protect against ovarian, endometrial, and colorectal cancer [22].

Another studied factor is BMI (Body Mass Index) [23]. In postmenopausal women with normal body mass index, relatively high body fat levels were associated with an elevated risk of invasive BC [24]. Regarding this factor, Mohanty et al. [2] demonstrated that premature menopause and premenopausal obesity decrease the risk, whereas postmenopausal obesity amplifies the risk.

In [25], García-Esquinas et al. studied the association of diabetes and diabetes treatment with risk of postmenopausal BC. Results showed that diabetes was not associated with the overall risk of BC, and it was only linked to the risk of developing triple negative tumors.

In [26], Pastor-Barriuso et al. combined estrogenic effects of mixtures of xenoestrogens in serum and their relationship to BC risk. Results shown a strong positive association between serum total xenoestrogen burden and BC risk, highlighting the importance of evaluating xenoestrogen mixtures, rather than single compounds, when studying hormone-related cancers.

The relation between alcohol consumption and BC has been shown in several works [27,28,29]. However, the mechanism of alcohol-induced carcinogenesis is not fully understood yet.

Many other factors have been studied to determine its association with BC, like environmental factors, pregnancy, sex, physical activity, economic level, multitude of genetic information, etc. [30,31,32,33].

In the problem we are dealing with, which is predicting the risk of breast cancer, it is not only performance that matters. It is also important to extract the most relevant features in order to better understand the data and the entire underlying process.

The digital world of data is expanding, with an annual growth rate of 40%, and health care is among the fastest growing sectors of the digital world, with an annual growth rate of 48% [34].

Small changes in the data give rise to differences in the results of the classification algorithms, and for this reason, in recent years there have been studies that perform feature selection with respect to classification performance [35,36,37,38]. For this reason, the study of the stability of features selection techniques has gained more and more importance [39,40].

In this work, the problem of calibrating a BC risk prediction model is addressed. Moreover, our focus is to extract the the most relevant factors applying feature ranking techniques on a real BC dasaset from the MCC-Spain study [41]. Both the risk prediction model performance and its stability are assessed jointly for this purpose.

In particular, we evaluate several feature ranking techniques in the context of breast cancer prediction. From the complete data set, it is proposed to extract different subsamples of data. A classification technique is applied to each of these subsamples, and this leads us to different rankings of features. Finally, the feature selection method is evaluated in relation to the performance of the classifier and in relation to its stability. The main contributions of this work are:An evaluation of multiple ranking feature methods has been carried out to extract the most relevant factors of breast cancer.All the feature selection methods have been evaluated, taking into account not only the performance, but also their stability.A deeper evaluation of the risk of breast cancer in menopausal status has been done by calibrating the risk prediction for pre- and post-menopausal women.Our automatically selected features have been compared with expert selected features in order to validate the algorithm results.Our method based on environmental or genetic characteristics helps to improve the quality of life of people suffering from the disease, thanks to an early diagnosis due to not having to wait for the clinical characteristics to appear.

The rest of the paper is organized as follows: Section 2 introduces the methodology used in this study. Results on a representative breast cancer dataset are presented in Section 3. A deeper analysis of the data, taking into account the menopausal status of the women, is discussed in Section 4. Section 6 lists the conclusions of this work and presents future works in this field.

## 2. Methods

Feature selection methods assess the relevance of a feature or a set of features according to a given measure. These techniques may provide many benefits, the most important ones being [42]: (a) to prevent overfitting and improve model performance, (b) to gain a deeper insight into the problem, and (c) to provide faster and more cost-effective predictive models.

Consider each sample $x_{i}$, defined in a *p*-dimensional vector $x_{i}=\left(x_{i1},x_{i2},\ldots x_{ip}\right)$ where each component $x_{ij}$ represents the value of a given feature $f_{j}$, for that example *i*, that is, $f_{j}\left(x_{i}\right)=x_{ij}$.

Consider also the training dataset $D=\{\left(x_{i},d_{i}\right),i=1,\ldots ,M\}$ with M examples and a class label *d* associated with each sample.

From a functional point of view, the output of a feature selection algorithm may be a ranking (weighting-score) on the features or a feature set. Obviously, representation changes are possible and, thus, a feature subset can be extracted from a full ranked list by selecting the most relevant features.

Consider now a feature ranking algorithm that provides a ranking vector r with components defined in (Equation 1)
(1)$$r=(r_{1},r_{2},r_{3},\ldots ,r_{p})$$
where $1\leq r_{i}\leq p$. Note that 1 is considered the highest rank. Consider also a feature subset (as denoted in (Equation 2)) with *k* elements as the outcome of a feature selection technique
(2)$$s=\left(s_{1},s_{2},s_{3},\ldots ,s_{p}\right),s_{i}\in \left\{0,1\right\}$$
where 1 indicates the presence of a feature and 0 the absence and $\sum_{i=1}^{p}s_{i}=k$ for a top-k list.

From a structural point of view, feature selection methods can be categorized into three groups: *filter*, *wrapper* and *embedded* approaches [36,43,44]. The *filter* techniques rely on general characteristics of the training data to rank the features according to a metric being independent of the classifier. The *wrapper* approaches select candidate subsets of features and assess their fitness based on the classification model performance. Finally, in the *embedded* techniques, the feature search mechanism is incorporated into the classifier objective function and are, therefore, specific to a given inductive learning algorithm. The ranking methods assessed in this work are briefly described next.

### 2.1. Feature Selection with Filters

Relief and Pearson algorithms have been considered. Relief is a well-known technique sensitive to feature interactions, and Pearson is a parameter-free approach that has proven to be very effective, although it does not eliminate redundancy [45].

#### 2.1.1. Relief

The main idea of the Relief algorithm is to calculate a feature score for each feature that can then be applied to rank and select the highest scoring features for feature selection. Alternatively, these scores can be applied as feature weights to guide subsequent modeling. The relief feature scoring is based on identifying the feature value differences between the pairs of nearest neighbor instances. If a characteristic value difference is observed in a pair of neighboring instances with the same class, the feature score decreases. Alternatively, if a characteristic value difference is observed in a pair of neighboring instances with different class values, the characteristic score increases [46].

#### 2.1.2. Pearson Correlation Coefficient

Pearson’s correlation coefficient is a test that measures the statistical relationship between two continuous variables. In this case, between each feature and the classes labels. If the association between the elements is not linear, then the coefficient is not adequately represented.

The correlation coefficient can take a range of values from +1 to −1. A value of 0 indicates that there is no association between the two variables. A value greater than 0 indicates a positive association. That is, as the value of one variable increases, so does the value of the other. If a feature is highly correlated with a class label, then this feature is relevant for our classification [46].

### 2.2. Feature Selection with Wrapper Approaches

Wrapper techniques rely on the performance of a learning algorithm to assess the importance of a feature set. They work by iteratively removing the least relevant features or by adding the most relevant features according to model performance [43]. Their main advantage is that they can select high-quality feature subsets for a particular classifier. Wrapper approaches are the methods with most computationally cost, though.

In this work, we evaluate two wrapper approaches that quantify the importance of a feature set based on the performance of a Support Vector Machine an a Logistic Regression classifier. In both cases, model performance is estimated by the area under the ROC (Receiver Operating Characteristic) curve (AUC), where the ROC curve plots the true positive rate against the false positive rate.

### 2.3. Feature Selection with Embedded Approaches

Two embedded approaches have been evaluated in this work: Random Forests and Support Vector Machines. These techniques based on embedded approaches have the advantage that they are better in terms of computational cost and also they have proven to be effective in terms of classification performance.

#### 2.3.1. SVM with Recursive Feature Elimination (SVM-RFE)

SVM-RFE [43] is based on SMV algorithm. This approach is capable of determining which are the most representative features to the model predictive power while it is being created.

#### 2.3.2. Random Forests (RF)

Random Forests is a combination of predictor trees such that each tree depends on the values of a random vector tested independently and with the same distribution for each of these. It is a substantial modification of bagging that builds a long collection of uncorrelated trees and then averages them. Each node of the multiple decision trees represents a condition over a single feature. Using this information, a ranked feature list can be extracted [43].

### 2.4. Stability of Feature Ranked Lists

Feature selection methods are used to measure the importance of a particular feature or group of features taking into account the value of a particular function. The field of cancer risk prediction certainly benefits from the identification of the most important variables (features) in order to better understand the data and the underlying process. A problem that appears in many practical problems, in particular when the available dataset is small and the feature dimensionality is high, is that small variations in the data lead to different outcomes of the feature selection algorithm.

In feature selection methods, it is important to take stability into account since, if the result of the technique varies under small changes in the data, then the conclusions to be drawn from it are not reliable [47,48,49].

If a feature ranking algorithm is ran *K* times, the Results can be represented in a matrix A with elements $r_{ij}$ with i=1,…,p and j=1,…,K that indicate the rank assigned in the run-*j* for feature-*i*. The same applies to a feature selector (Equation (Equation 3)).
(3)$$A=\left[\begin{array}{ccccc} r_{1} & r_{2} & \;\ldots \; & r_{j} & r_{K} \end{array}\right]={\left[\begin{array}{ccccc} r_{11} & r_{12} & \;\ldots \; & r_{1j} & r_{1K}\\ r_{21} & r_{22} & \;\ldots \; & r_{2j} & r_{2K}\\ r_{31} & r_{32} & \;\ldots \; & r_{3j} & r_{3K}\\ r_{i1} & r_{i2} & \;\ldots \; & r_{ij} & r_{iK}\\ \;\ldots \; & \;\ldots \; & \;\ldots \; & ... & \;\ldots \;\\ r_{p1} & r_{p2} & \;\ldots \; & r_{pj} & r_{pK} \end{array}\right]}_{p\times K}$$

In recent years, numerous investigations have appeared focused on evaluating the stability of the characteristics selection methods, especially when one wants to obtain information from the data taking into account the most relevant features [39,40,50,51,52,53,54,55].

Stability is typically quantified by calculating the pairwise similarity of a set of classifications and then downgrading them to a single metric. These metrics can be projected in one-dimensional space. In this article, we propose the use of graphical methods to also evaluate the stability of the classifiers making projections in two dimensions.

Finally, a conventional analysis and also a graphical analysis will be carried out to quantify the robustness of the feature selection method.

#### 2.4.1. Conventional Stability Analysis

Assuming that the algorithm is launched *K* times on slightly different datasets extracted from the training dataset. Then, a set of outputs from a feature ranking algorithm represented as $A=\{r_{1},r_{2},\ldots r_{K}\}$ is obtained. A single scalar value can be obtained by evaluating the stability of the set by calculating the pairwise similarities and then averaging the results.
(4)$$S\left(A\right)=\frac{2}{K(K-1)}\sum_{i=1}^{K-1}\sum_{j=i+1}^{K}S_{M}\left(r_{i},r_{j}\right)$$
where $S_{M}$ may be any distance metric such as the Spearman rank correlation coefficient, Jaccard stability index [50,56] or Kuncheva’s stability index [57].

Other alternatives compute the stability directly from the whole set of lists without carrying out pairwise comparisons [48].

In this paper, several metrics are proposed to study the stability of the feature ranking or selection techniques.

##### Similarity Metrics for Feature Rankings

Consider r and $r^{\prime }$ the output of a feature ranking technique applied to two subsamples of D. The Spearman’s rank correlation coefficient (SR) is the most popular metric to compare the similarity between two rankings [56]. The SR between two ranked lists r and $r^{\prime }$ is defined by
(5)$$SR\left(r,r^{\prime }\right)=1-6\sum_{i=1}^{p}\frac{{\left(r_{i}-r_{i}^{\prime }\right)}^{2}}{p(p^{2}-1)}$$
where $r_{i}$ is the rank of feature-*i*. SR values range from −1 to 1. It takes the value of one when the rankings are identical, and the value zero when there is no correlation.

##### Similarity Metrics for Feature Selectors

When we attempt to measure the distance between two top-k lists *s* and $s^{\prime }$ with the most relevant k features, several metrics have been presented (for details see [56]). In this work, we use the Jaccard stability index (JI) that can be defined as
(6)$$JI\left(s,s^{\prime }\right)==\frac{s\wedge s^{\prime }}{s\vee s^{\prime }}=\frac{r}{l}$$
where *s* and $s^{\prime }$ are the two feature subsets, *r* is the number of features that are common in both lists and *l* the number of features that appear only in one of the two lists. The JI lies in the range (0,1).

#### 2.4.2. Stability Graphical Analysis

When we talk about a classification algorithm, the result is interpreted as a point within a high-dimensional space. In these cases, the stability of the method is measured as the distance between different results of the same range averaging the results. In this way, projecting the data to a single dimension, it becomes a single number and can now be compared with a scalar metric. The only limitation is that in this case, it is only possible to compare the feature selector with respect to a reference: The random classification and the completely stable classification.

If we carry out a projection in two dimensions, we can compare with respect to the random selector, but we can also compare each selector of characteristics with the others, therefore, it is a better option.

Evaluating different rendering techniques, histograms and scatter plots, which are very simple visualization methods, have some limitations when increasing dimensionality. For this reason, we think it is a better idea to use MultiDimensional Scaling (MDS) [58], as it preserves most of the original data structure. This technique allows multidimensional data to be projected in a two- or three-dimensional space and also preserves the distances of the original multidimensional space. The first time this technique was used was to compare classifiers against multiple metrics, in the field of machine learning [59].

## 3. Experimental Results: Breast Cancer Dataset

In this paper, a model for predicting the risk of suffering from breast cancer is proposed by evaluating different algorithms for feature selection. To evaluate all these algorithms we have relied on both the performance of the classifiers and the robustness of the ranking algorithms.

### 3.1. Breast Cancer Dataset

Experimental results were carried out using a BC dataset obtained from the MCC-Spain study [41]. MCC-Spain is a multicentric case–control study with population controls aiming to evaluate the influence of environmental exposures and their interaction with genetic factors in common tumors in Spain (prostate, breast, colorectal, gastroesophageal and chronic lymphocytic leukemia). All participants signed an informed consent. Approval for the study was obtained from the ethical review boards of all recruiting centers [60].

For each individual, 124 features are considered:Fifty environmental factors including red meat, vegetable consumption, BMI, physical activity, alcohol consumption, etc.;64 genetic variables (Single Nucleotide Polymorphisms -SNPs);Other variables (10) such as family history of BC, age or education level.

A preprocessing of the data has been carried out by eliminating those that had missing values, and after this process, we have a data set of 1865 instances. Of those, 946 are controls and 919 are cases. The variables that have been taken into account in this research are the following:Environmental factors: phenols, oil, oral contraceptives, NSAIDs *(nonsteroidal anti-inflammatory drugs)*, BMI, red meat, cereals, flavonoids, fruits, smoker, dairy, legumes, lignans, ethyl alcohol, fish, stilbenes, calcium, carotenoids, cholesterol, edible, total energy, total fats, dietary fiber, folic acid, total carbohydrates, iron, magnesium, phosphorus, polysaccharides, potassium, animal proteins, vegetal proteins, total proteins, retinoids, sodium, digestible sugars, zinc, A-group vitamins, vitamin B1, vitamin B12, vitamin B2, vitamin B3 *(niacin)*, vitamin B6, vitamin C, D-group vitamins, E-group vitamins, HRT *(Hormone Replacement Therapy)*, vegetables, Mets *(metabolic equivalent of task)* in 10 years and QPA *(Quality of Physical Activity)*.SNP: rs-1042522, rs-11085147, rs-137902538, rs-138607522, rs-139554429, rs-139697494, rs-141143854, rs-141363120, rs-141420305, rs-142068825, rs-143582231, rs-144811392, rs-145519500, rs-145760222, rs-146208471, rs-146505192, rs-146848959, rs-146875699, rs-1470383, rs-147307965, rs-148214998, rs-148728256, rs-149210226, rs-149633775, rs-150378600, rs-17187428, rs-17880282, rs-190372148, rs-199803800, rs-200147790, rs-200239262, rs-200431478, rs-201029843, rs-201100551, rs-201340741, rs-201498076, rs-201652303, rs-201664019, rs-201686188, rs-202004587, rs-202041676, rs-2230461, rs-2279744, rs-2287498, rs-2287499, rs-2758331, rs-2855116, rs-34154613, rs-34402166, rs-35804229, rs-36084391, rs-3730581, rs-3824120, rs-4516970, rs-4645956, rs-4645959, rs-4645961, rs-4726020, rs-4880, rs-5746096, rs-5746105, rs-71310379, rs-78419579 and rs-937283.Other factors: family history of BC, age, education level, offspring, menopausal, nulliparous, abdominal obesity, age of menarche, lactation months and socioeconomic level.

### 3.2. Predictive Power Assessment

In order to predict the BC risk, many different models have been evaluated: Logistic Regression, k-Nearest Neighbors, Neural Networks with a Multilayer Perceptron architecture, Support Vector Machines and Boosted Trees. We have decided to include all of these methods because they have been successful in a wide variety of fields of study, as shown in the state of the art evaluation. The best hyperparameters were chosen experimentally using a Grid Search methodology, and the configuration is detailed below.

Logistic Regression (LR). Logistic regression classifier was trained using liblinear solver, regularization factor C = 1 and 1000 iterations.

k-Nearest Neighbors (k-NN). Nearest neighbors with k = 21 are extracted using the Minkowski distance, which is a generalization of Euclidean and Manhattan distances. Features are normalized with mean equals to zero mean and standard deviation equals to one.

Support Vector Machines (SVM). SVM with a radial basis function kernel has been used. The training step was performed using the Sequential Minimal Optimization routine.

Boosted Trees (BT). The AdaBoost-SAMME ensemble aggregation method was used. For these experiment, we selected a learn rate of 1. The maximal number of estimators was set to 500.

Multilayer Perceptron (MLP). We evaluate a three layer neural network with 200 neurons in the hidden layer and a logistic sigmoid activation for all the layers. The network has been trained using the adam optimizer with a maximum number of 5000 iterations.

The classification performance (AUC) is estimated using 7-fold cross validation.

To obtain a reference value, Table 1 shows the AUC of the classifiers used on the entire feature set, without performing any feature selection process.

Feature selection has been carried out in order to train the models with the most relevant features. Six feature rankers have been used: Two of them based on a *filter approach* (ReliefF and the Pearson correlation coefficient), another two follow a *wrapper approach* (SVM and Logistic Regression guided by the AUC classifier performance with 7-fold cross validation) and two are *embedded approaches* (SVM-RFE and RF).

Data are normalized to zero mean and unit standard deviation. The ranking algorithm is run several times with 70 % of data, randomly extracted from the entire dataset. Seven runs of this process resulted in a total of seven different rankings for each feature selection technique. The ranking that has been used for this purpose is the average of the seven rankings that were obtained by executing the algorithm seven times. All method programming has been done using Python as the programming language.

In Figure 1, Figure 2, Figure 3, Figure 4 and Figure 5, we can see the AUC of five different classifiers (AdaBoost, k-NN, Logistic regression, MLP and SVM) trained with a number of features ranging from 1 to 124. These features were selected taking into account the relevance of the six classification algorithms that are evaluated in this article. Classifier performance can also be observed using the full feature set without performing feature reduction techniques. k-NN and AdaBoost classifier seem to be more unstable, but in the rest of the models, it can be seen that performance increases as we increase the number of features used as predictors. However, as new features are added, you can see how SVM, MLP and LR perform worse. This is because those new features that are added are the most irrelevant or even redundant.

As can be seen, the highest AUC for MLP is 0.615 selecting the 46 most important characteristics using the SVM-RFE ranker. If we use 124 characteristics, the AUC is reduced to 0.579 (see Figure 2). For LR and also using the SVM-RFE ranker, it can be observed that the AUC increases from 0.578 if all the features are used, to a value of 0.616 if the most relevant 46-49 ones are used (Figure 3).

As the analysis is very complicated, we can observe Table 2, Table 3, Table 4, Table 5 and Table 6 where the best feature selection strategy has been included, for each classifier, using as a metric area under the curve (AUC). This is carried out for the top-30, top-60 and top-90 features. Thus, Table 2, Table 3, Table 4, Table 5 and Table 6 collect the three best feature sets up to a cardinality of 90 features that lead to the best performance for each one of the classifiers and feature selection techniques.

In these tables, we can see that if we consider the 90 most relevant features, the LR technique works better with: top-46 SVM-RFE, top-47 SVM-RFE and top-49 SVM-RFE. It seems clear that the ranker that offers the best results in most cases is SVM-RFE.

Best feature selection techniques for Top-30, Top-60 and Top-90 lists are shown in Table 2, Table 3, Table 4, Table 5 and Table 6.

### 3.3. Ranking Stability Analysis

Seven runs of each feature raking algorithm have been carried out. This results in seven different classifications. Feature raking algorithms have been released with 70 % of the total set data randomly drawn.

#### 3.3.1. Traditional Stability Analysis

There are metrics such as the Spearman’s rank correlation coefficient (rho), with which we can evaluate the stability of the feature ranking algorithms. The $\frac{7(7-1)}{2}$ pairwise similarities for each algorithm have been computed to end up averaging these computations according to Equation (Equation 4). The ρ value is shown in Table 7 where it can be seen that SVM-RFE is the most stable (0.474) ranking algorithm, and LR-Wrapper is quite unstable (0.030).

Another technique we have used is the Jaccard index, with which the stability of a subset of features that contains the top- *k* feature lists can be studied. In Table 8, the Jaccard index for the selection of feature subsets with cardinality that varies from 10 to 124 and the average in the last row is shown. In view of these results, we can say that the wrapper approaches are not very stable, and that the embedded approaches are. If we look at both the performance and the stability of the classifier, it can be seen that RF is the most stable technique, but the performance is not as good as other classifiers. On the other hand, SVM-RFE has a moderate robustness, but it is the best classification technique in view of the results.

If the analysis is based on a single metric, we have no way of telling how similar the rankings provided by the different algorithms are. The questions we should be able to answer are, firstly, which classifiers provide similar classifications and secondly, which classifier is more stable for a certain range of *k* values. Results shown in Table 8 are not easy to interpret.

#### 3.3.2. Graphical Stability Analysis

In Figure 6a it can be seen that the relative stability changes as a function of the value of *k*. In general, it appears that the RF and Pearson algorithms show the most stable results, and the stability of SVM-RFE is very low at low *k* values. It is not possible to extract the most relevant features if the algorithm is run only once. To obtain a more representative ranking, a good option is to add the rankings. Likewise, wrapper approaches are very unstable.

To view and compare the feature selectors, we use MDS [58].

The experiments that have been carried out can be represented as a set of 46 points since we have used six algorithms and each algorithm has been launched seven times. In this way, a 124-dimensional space is defined. Using the MDS, these points are projected in two dimensions. The distance between points is calculated with the Spearman’s rank coefficient, and the stress criterion is normalized with the sum of squares of the dissimilarities.

After making this representation in two dimensions, each result of the algorithm is represented by two coordinates (x,y). In Figure 6b, the similarities between the feature selector can be seen. In terms of stability, it can be seen that the Pearson, ReliefF and LR-wrapper points are scattered while those of SVM-RFE, SVM-wrapper and Random Forest are more clustered.

In addition to studying the stability of the methods, the ability to predict the correct class also has to be studied together. This is important, as experts need information on the most relevant risk factors and protective factors, and not just information on which methods are more stable. In terms of predictive power (see Figure 1 in previous sections), MLP and LR shows the best behavior when they are fed with SVM-RFE ranking. This is also confirmed with the analysis conducted in Section 3.2 (Table 5 and Table 6).

### 3.4. Comparison with the State-of-the-Art Knowledge

Several recent works address the problem of BC prediction [2,21,31,61,62,63]. In this section, the performance of a model will be evaluated using a data set consisting of 19 features that have been selected by experts in this field.. These features (see Table 9) are: 7 SNP (Rs-146875699, rs-2279744, rs-190372148, rs-71310379, rs-137902538, rs-202004587 and rs-149633775), 7 environmental features (NSAIDs, contraceptives, BMI, smoker, total energy, Mets and alcohol) and 5 other variables (Family history of BC, age, offspring, socioeconomic level and age of menarche).

In addition, three (“Top-47 SVM-RFE”, “Experts’ set ∪ Top-47 SVM-RFE” and “Experts’ set ∩ Top-47 SVM-RFE”) more feature sets have been created through our experimental work (see Section 3.2 for further information about how the lists of features were experimentally built. Section 2 describes the different feature selection techniques applied), based on different combinations of the features listed in Table 9 (experts’ features) and Table 10 (experimentally most relevant features).

Then, these four feature subsets, jointly with the full feature set, was assessed in terms of AUC by every classifier considered in this study. Table 11 shows the performance provided by the classifiers for each feature set.

When the number of features is lowered from 19 (*experts’*) to 10 (*Experts’ set ∩ Top-47 SVM-RFE*), the performance measurement (AUC) just slightly increases for most of the classifiers, which results in almost doubling the importance of each feature.

If the features that our method does not consider relevant are eliminated, it can be seen in Table 11 that the AUC increases or is maintained. Thus, AUC for the LR classifier increases from 0.582 to 0.616 (about +5.89%) and from 0.580 to 0.611 (about +5.38%) for the SVM approach comparing the full feature set with the experimentally 47 most relevant features (about −62% of features).

With this analysis we can conclude that some of the features categorized by the experts as important are actually irrelevant, since performance is not affected by disregarding them. This is also confirmed by the fact that some of these features do not occupy the first positions in the ranking lists obtained in our experimental environment. This is the case of “offspring”, “socioeconomic level”, “BMI”, “age of menarche”, “METS” and some SNPs.

It is also important to note that if the AUC obtained using the features provided by the experts is compared with the complete set, a slight increase is seen for some classifiers. This indicates that some features not included in the list are more relevant.

## 4. A Deeper Data Analysis

Analyzing the risk of BC considering the menopausal status becomes an interesting issue. We calibrate a risk prediction model for women with pre- and post-menopausal status in Section 4.1 and Section 4.2, respectively.

### 4.1. Pre-Menopausal

Pre-menopausal data partition consists of 1233 samples (from which 569 are cases and 664 are controls) and each sample is composed of 123 features. Before performing any feature selection, the five classification techniques achieve the performance shown in Table 12. As we can see, the best results are achieved using SVM classifier with a 0.575±0.032 followed by LR with a 0.572±0.024.

Once the feature selection rankings are calculated and iterative performance is measured for each number of features, the best performance (0.588±0.037) occurs with top-52 features of SVM-RFE ranking and LR classifier (see all performances in Figure 7). Feature selection using SVM-RFE method in pre-menopausal breast cancer data classification shows a better performance with all the classifiers. As we can see, with LR, MLP and SVM, the best AUC is obtained with reductions greater than 50% of the raw data, whereas BT and KNN show unstable results along the cardinality of the feature subset.

Top-30 SVM-RFE features can be shown in Table 13. To create the ranking, median has been taken into account to avoid the influence of the extreme values. As we can see, features like Phenols, Nulliparous, Legumes, METS in 10 years and QPA seems to be the most relevant ones for this subset experimentation.

### 4.2. Postmenopausal

The postmenopausal subset of data is made of 632 samples (from which 350 are cases and 282 are controls) and each sample is characterized by 123 features. Before performing any feature selection, the five classification techniques achieve the performance shown in Table 14. In this case, kNN method outperforms all the other models with a 0.632±0.068 AUC followed by SVM with a 0.594±0.069.

Once the feature selection rankings are calculated and iterative performance is measured for each number of features, the best performance (0.632±0.068) occurs with top-122 features of RFE ranking and k-NN classifier (see all performances in Figure 8). As we can see, with postmenopausal data, the feature reduction does not affect positively in the AUC performance in any of the classifiers using SVM-RFE ranking.

Although the reduction in characteristics does not increase the AUC, the ranking continues to provide relevant information, the features are sorted according to their importance in the classification of BC. Top-30 SVM-RFE features for postmenopausal data partition can be shown in Table 15. To create the ranking, median has been taken into account to avoid the influence of the extreme values. In this case, Phenols, BMI, QPA, Cereals and different vitamins (B2, C, B1, E, B12, A and D) shows to be relevant in this experiment.

## 5. Clinical Practice Points

Machine learning models can be calibrated to predict which people are at higher risk of developing breast cancer what facilitates earlier identification and intervention.This helps to improve the quality of life of people suffering from the disease, thanks to an early diagnosis.This study based on intelligent systems identifies some relevant factors already related to breast cancer. Other factors (SNPs and environmental factors such as legumes intake and blood iron level) also turned out to be relevant. Further study is, however, needed before they are included as suspected risk factors for breast cancer.

## 6. Conclusions and Future Work

The aim of this work is to identify the most relevant features regarding BC prediction. At the same time, this helps to improve the BC prediction model performance. Experimental results showed that the best performances are achieved with a LR classifier (AUC = 0.616) using the top-47 features selected by the SVM-RFE approach and a MLP neural network (AUC = 0.615) using the top-46 features selected also by SVM-RFE. With this, it can be concluded that there is an improvement over the use of the full dataset of 5.84% and 7.71% for the LR and MLP classifier, respectively. A major advantage of SVM-RFE is that it can select high-quality feature subsets for this particular classification task. Despite the fact that it does not take into account any correlation the features might, it outperforms other feature ranking approaches.

Table 10 shows the top-47 variables extracted with SVM-RFE feature selection algorithm in comparison with the selection made by the experts. It can be seen that some features are common in both lists. They have been highlighted in gray. Thus, 10 out of 19 features selected according to the state of the art knowledge also appear in the automatic selection carried out by the SVM-RFE approach. One of the most important features is *age*, which is the most relevant selected by SVM-RFE.

In contrast, only 2 of the 10 most important features extracted by SVM-RFE are included in the experts’ selection (Age and Family history of BC). We would like to highlight the relevance of the other 8 features based on the improvements achieved in the classification step (5.30% using LR and a 5.49% with MLP classifier) and consider these factors deserve further research.

Stability is also assessed in this work with a scalar metric and also with a graphical approach. This graphical approach based on a MDS projection allows us to see easily that: (a) the most stable algorithms are SVM-RFE and Random Forest and (b) Pearson, Relief, SVM-wrapper and LR-wrapped are very unstable.

The main strength of our proposal is that the stability and predictive power of the models are analyzed at the same time. In addition, it is possible to identify the features that most influence breast cancer. In this study, it is concluded that SVM-RFE is one of the best techniques considering performance and robustness. This stability reinforces the reliability of the feature ranking derived with SVM-RFE.

A comprehensive evaluation with more data on BC and more variables is proposed as feature work so that we can re-confirm our findings and also reach more generalization. It would also be interesting to include the study of ensemble strategies to increase the stability of feature selection techniques. Because classification methods tend to be computationally expensive when dealing with large amounts of data, we aim to use hybrid classification methods based on two steps. First, the use of filters to quickly remove irrelevant features and second, wrapper or embedded ranking algorithms focused on the subset of features selected in the first step. It would be also interesting to use some reduction techniques such as PCA [64] or LDA [65] before ML training.

Furthermore, an evaluation of the explicability of all the proposed methods would be very interesting for the selection of characteristics in order to improve the acceptability of the study.

## Figures and Tables

**Figure 1 ijerph-18-10670-f001:**
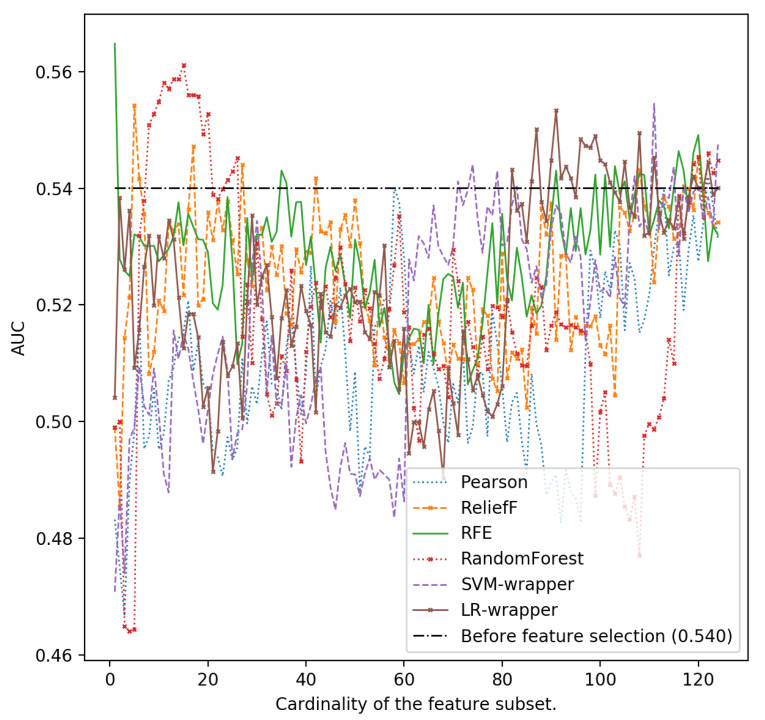
Area under the curve using the complete data set without reducing features and different cardinality of the subset of features for different classifiers: AdaBoost.

**Figure 2 ijerph-18-10670-f002:**
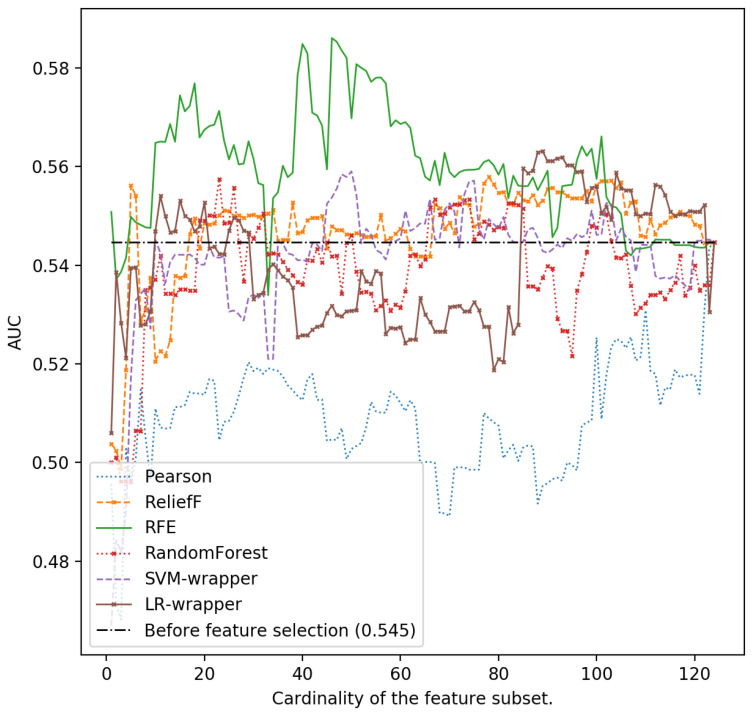
Area under the curve using the complete data set without reducing features and different cardinality of the subset of features for different classifiers: k-NN.

**Figure 3 ijerph-18-10670-f003:**
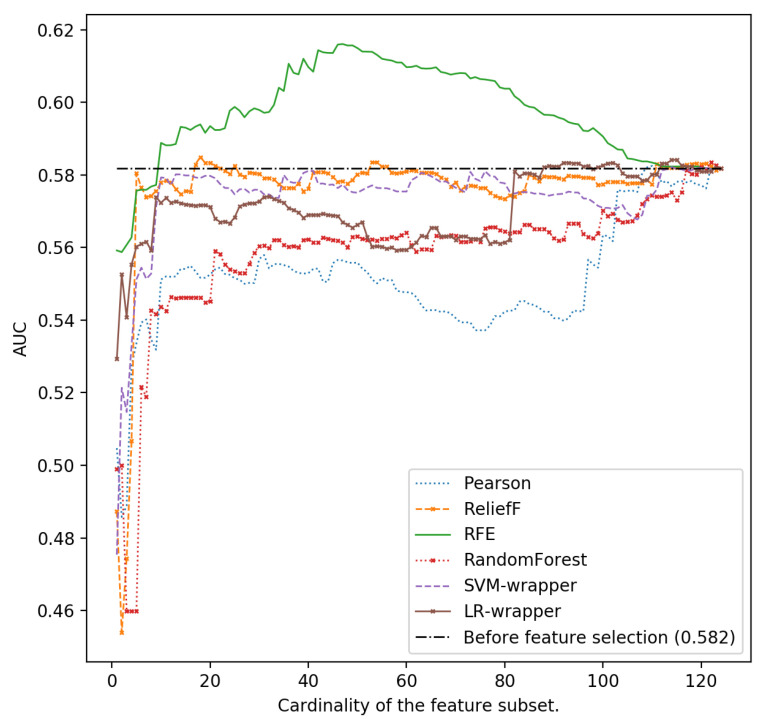
Area under the curve using the complete data set without reducing features and different cardinality of the subset of features for different classifiers: Logistic Regression.

**Figure 4 ijerph-18-10670-f004:**
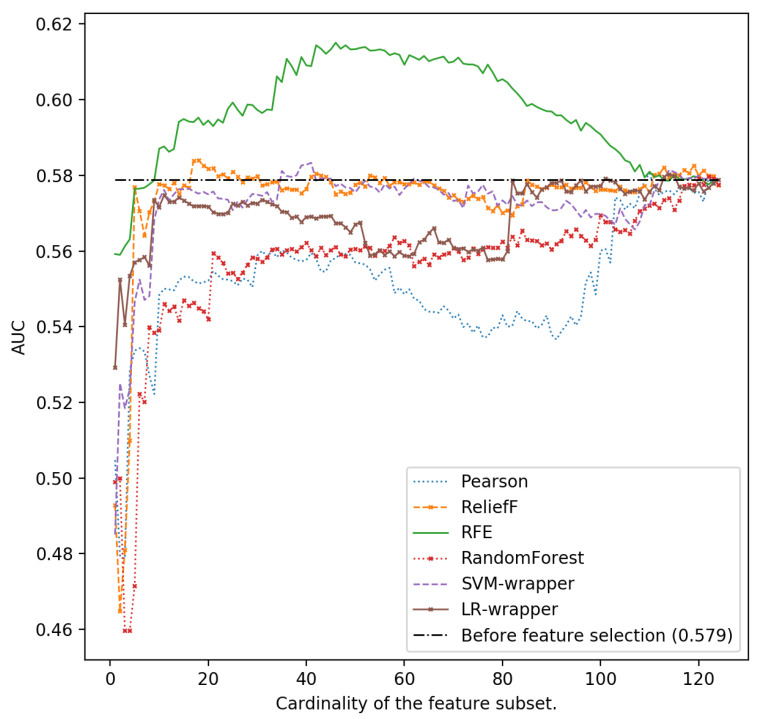
Area under the curve using the complete data set without reducing features and different cardinality of the subset of features for different classifiers: Multilayer Perceptron.

**Figure 5 ijerph-18-10670-f005:**
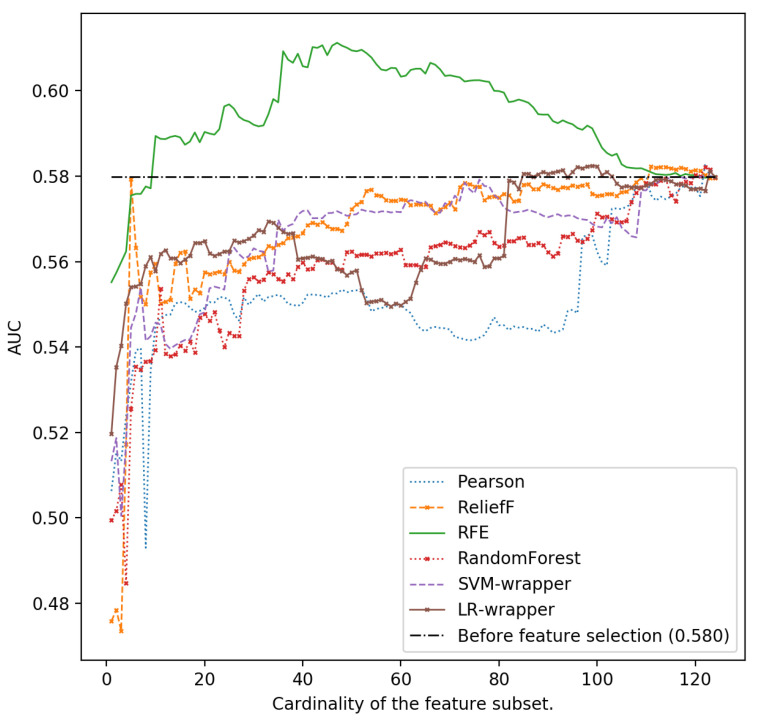
Area under the curve using the complete data set without reducing features and different cardinality of the subset of features for different classifiers: SVM.

**Figure 6 ijerph-18-10670-f006:**
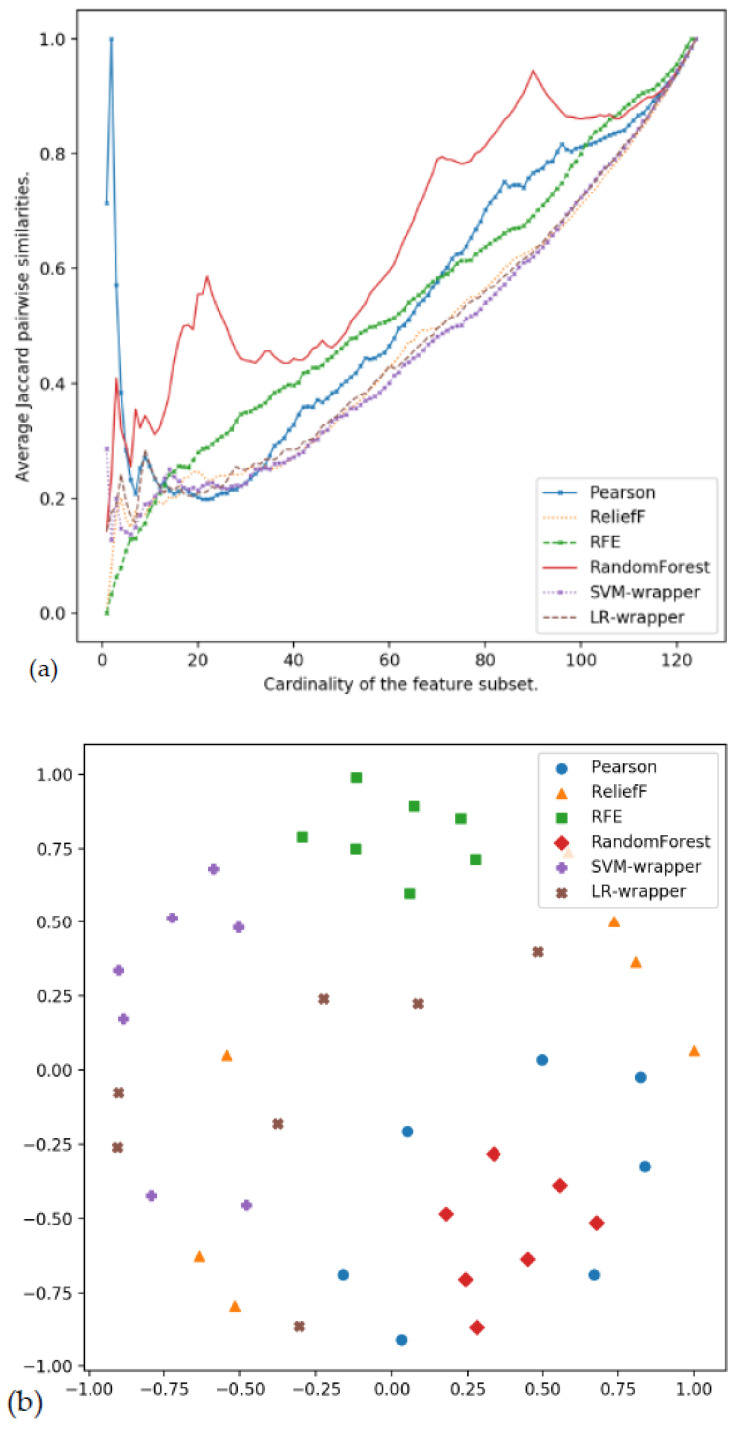
Feature selector stability: (**a**) Jaccard index for feature subsets with different cardinality; (**b**) MDS plot of the feature ranking algorithms.

**Figure 7 ijerph-18-10670-f007:**
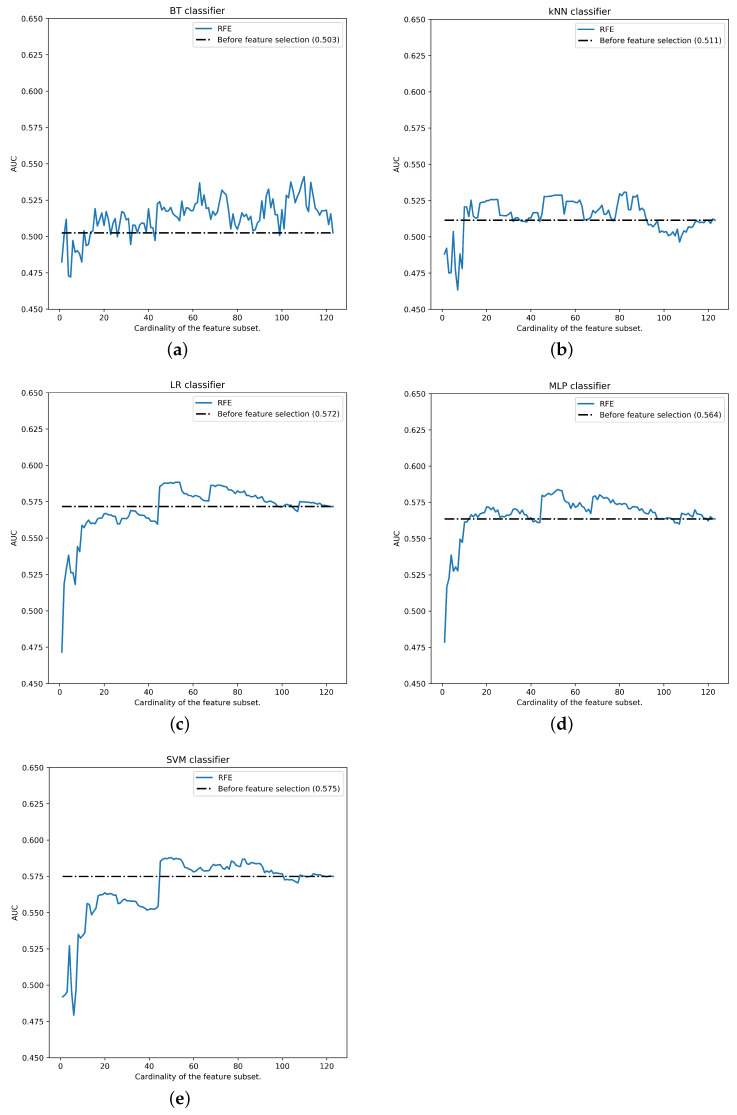
Performance for pre-menopausal data partition (by classifier): (**a**) BT, (**b**) k-NN, (**c**) LR, (**d**) MLP, (**e**) SVM.

**Figure 8 ijerph-18-10670-f008:**
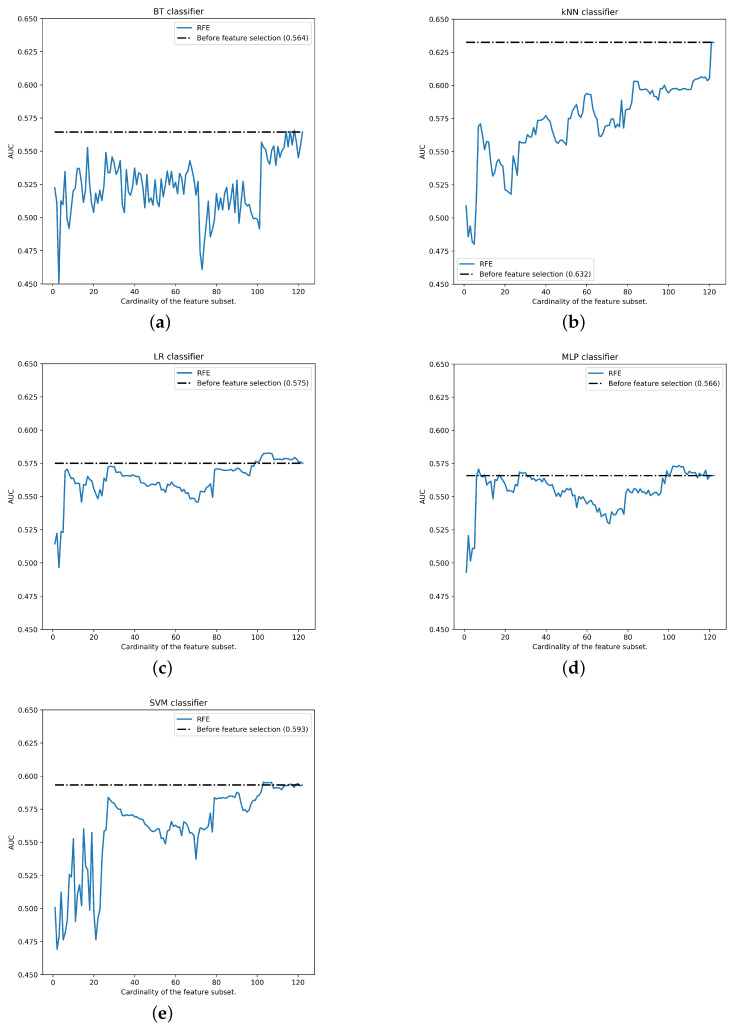
Performance for postmenopausal data partition (by classifier): (**a**) BT, (**b**) k-NN, (**c**) LR, (**d**) MLP, (**e**) SVM.

**Table 1 ijerph-18-10670-t001:** AUC for different classifiers with the original dataset (without performing feature selection).

LR	SVM	MLP	k-NN	BT
0.582	0.580	0.578	0.545	0.544
(±0.039)	(±0.040)	(±0.041)	(±0.044)	(±0.021)

**Table 2 ijerph-18-10670-t002:** SVM classifier. In gray background the number of features with the best result.

	#Features	Ranking	AUC
**Top-30**	24	SVM-RFE	0.596 (±0.040)
	25	SVM-RFE	0.597 (±0.039)
	26	SVM-RFE	0.596 (±0.038)
**Top-60**	44	SVM-RFE	0.611 (±0.048)
	46	SVM-RFE	0.611 (±0.040)
	47	SVM-RFE	0.611 (±0.039)
**Top-90**	44	SVM-RFE	0.611 (±0.048)
	46	SVM-RFE	0.611 (±0.040)
	47	SVM-RFE	0.611 (±0.039)

**Table 3 ijerph-18-10670-t003:** Boosted Trees classifier. In gray background the number of features with the best result.

	#Features	Ranking	AUC
**Top-30**	1	SVM-RFE	0.565 (±0.034)
	13	RF	0.559 (±0.030)
	15	RF	0.561 (±0.032)
**Top-60**	1	SVM-RFE	0.565 (±0.034)
	13	RF	0.559 (±0.030)
	15	RF	0.561 (±0.031)
**Top-90**	1	SVM-RFE	0.565 (±0.034)
	13	RF	0.559 (±0.030)
	15	RF	0.561 (±0.031)

**Table 4 ijerph-18-10670-t004:** k-NN classifier. In gray background the number of features with the best result.

	#Features	Ranking	AUC
**Top-30**	15	SVM-RFE	0.575 (±0.035)
	17	SVM-RFE	0.572 (±0.038)
	18	SVM-RFE	0.577 (±0.038)
**Top-60**	40	SVM-RFE	0.585 (±0.047)
	46	SVM-RFE	0.586 (±0.040)
	47	SVM-RFE	0.585 (±0.041)
**Top-90**	40	SVM-RFE	0.585 (±0.047)
	46	SVM-RFE	0.586 (±0.040)
	47	SVM-RFE	0.585 (±0.041)

**Table 5 ijerph-18-10670-t005:** Logistic Regression classifier. In gray background the number of features with the best result. In blue background the best result over all the experiments.

	#Features	Ranking	AUC
**Top-30**	25	SVM-RFE	0.599 (±0.037)
	29	SVM-RFE	0.598 (±0.034)
	30	SVM-RFE	0.598 (±0.035)
**Top-60**	46	SVM-RFE	0.616 (±0.044)
	47	SVM-RFE	0.616 (±0.043)
	49	SVM-RFE	0.616 (±0.043)
**Top-90**	46	SVM-RFE	0.616 (±0.044)
	47	SVM-RFE	0.616 (±0.043)
	49	SVM-RFE	0.616 (±0.043)

**Table 6 ijerph-18-10670-t006:** Multi-Layer Perceptron classifier. In gray background the number of features with the best result.

	#Features	Ranking	AUC
**Top-30**	25	SVM-RFE	0.599 (±0.042)
	28	SVM-RFE	0.599 (±0.038)
	29	SVM-RFE	0.599 (±0.037)
**Top-60**	42	SVM-RFE	0.614 (±0.044)
	46	SVM-RFE	0.615 (±0.042)
	48	SVM-RFE	0.614 (±0.043)
**Top-90**	42	SVM-RFE	0.614 (±0.044)
	46	SVM-RFE	0.615 (±0.041)
	48	SVM-RFE	0.614 (±0.043)

**Table 7 ijerph-18-10670-t007:** Stability of a set with seven full rankings assessed by averaging pairwise similarities with the Spearman’s rank correlation coefficient (ρ). In gray background the best result.

	Pearson	ReliefF	SVM-RFE	Random Forest	SVM- Wrapper	LR- Wrapper
** ρ **	0.0179	0.052	0.474	0.411	0.066	0.030

**Table 8 ijerph-18-10670-t008:** Stability of a set with 7 top-k lists assessed through average pairwise similarities with the Jaccard index for different values of *k*. In gray background the best result.

k	Pearson	ReliefF	SVM-RFE	Random Forest	SVM- Wrapper	LR- Wrapper
10	0.256	0.188	0.179	0.329	0.192	0.192
20	0.202	0.247	0.280	0.555	0.214	0.214
30	0.227	0.247	0.350	0.440	0.226	0.226
40	0.328	0.272	0.397	0.443	0.272	0.272
47	0.374	0.316	0.441	0.466	0.320	0.320
50	0.397	0.344	0.460	0.479	0.343	0.343
60	0.466	0.428	0.511	0.596	0.401	0.401
70	0.577	0.495	0.584	0.790	0.482	0.482
80	0.702	0.565	0.638	0.814	0.541	0.541
90	0.767	0.635	0.692	0.945	0.622	0.622
100	0.812	0.711	0.800	0.861	0.724	0.724
110	0.850	0.812	0.887	0.875	0.822	0.822
120	0.942	0.939	0.957	0.943	0.945	0.945
124	1	1	1	1	1	1
Average for *k* from 1 to 124	0.545	0.474	0.541	0.654	0.471	0.480

**Table 9 ijerph-18-10670-t009:** Relevant features according to state-of-the-art knowledge. Features highlighted in bold are those that have also been found relevant in our study.

Relevant Features
rs146875699
**rs2279744**
rs190372148
**rs71310379**
rs137902538
**rs202004587**
rs149633775
**Age**
**NSAIDs**
Offspring
**Contraceptives**
Socioeconomic level
BMI
**Smoker**
**Total energy**
**Family history of BC**
Total Met in 10 years
**Ethyl alcohol**
Age of menarche

**Table 10 ijerph-18-10670-t010:** Top-47 SVM-RFE features and top expert selected features. Features highlighted are those that have also been found relevant in our study.

#	Top-47 SVM-RFE		SVM-RFE-Sorted Experts’	
	Feature	Median Rank Position	Feature	Median Rank Position
1	Age	6	Total energy	3
2	Legumes	6	Age	4
3	RS-201340741	8	RS-71310379	4
4	RS-141143854	9	NSAIDs	5
5	Family history of BC	10	Family history of BC	5
6	RS-201100551	10	RS-149633775	7
7	RS-148728256	12	RS-202004587	8
8	RS-146208471	13	>Oral contraceptives	9
9	RS-148214998	13	>Smoker (ever)	9
10	Iron	15	Ethyl alcohol	9
11	>RS-71310379	16	RS-146875699	9
12	RS-143582231	17	Offspring	12
13	RS-34154613	18	Socioeconomic level	12
14	RS-4645959	18	RS-137902538	12
15	Digestible sugars	19	RS-190372148	13
16	Folic acid	21	BMI	14
17	RS-78419579	24	Age of menarche	15
18	RS-144811392	25	METS in 10 years	16
19	RS-5746105	26	RS-2279744	16
20	Dairy	27		
21	Potassium	28		
22	RS-146848959	29		
23	Fish	31		
24	RS-2758331	31		
25	Vitamin C	31		
26	Vegetables	33		
27	RS-2287498	34		
28	Total energy	34		
29	Carotenoids	37		
30	Edible	38		
31	Fruits	39		
32	RS-202004587	39		
33	Flavonoids	40		
34	NSAIDs	41		
35	RS-138607522	42		
36	Abdominal obesity	43		
37	Ethyl alcohol	43		
38	RS-145519500	44		
39	Stilbenes	44		
40	RS-2279744	46		
41	Animal protein	46		
42	Oral contraceptives	47		
43	RS-141363120	50		
44	Magnesium	50		
45	Smoker (ever)	51		
46	Lignans	52		
47	Retinoids	54		

**Table 11 ijerph-18-10670-t011:** AUC for different classifiers with different feature sets. In gray background the best result is shown.

Feature Set	Cardinality	LR	SVM	MLP	k-NN	BT
Full feature set	124	0.582 (±0.039)	0.580 (±0.040)	0.578 (±0.041)	0.545 (±0.044)	0.544 (±0.021)
Experts’ set ∪ Top-47 SVM-RFE	57	0.612 (±0.046)	0.602 (±0.044)	0.609 (±0.045)	0.559 (±0.034)	0.523 (±0.038)
Top-47 SVM-RFE	47	0.616 (±0.043)	0.611 (±0.039)	0.614 (±0.044)	0.585 (±0.041)	0.517 (±0.039)
Experts’ set	19	0.585 (±0.044)	0.571 (±0.039)	0.582 (±0.040)	0.527 (±0.031)	0.545 (±0.041)
Experts’ set ∩ Top-47 SVM-RFE	10	0.587 (±0.040)	0.572 (±0.028)	0.585 (±0.037)	0.560 (±0.040)	0.544 (±0.051)

**Table 12 ijerph-18-10670-t012:** AUC for different classifiers with the pre-menopausal data partition (without performing feature selection).

LR	SVM	MLP	k-NN	BT
0.572	0.575	0.564	0.511	0.515
(±0.024)	(±0.032)	(±0.026)	(±0.060)	(±0.036)

**Table 13 ijerph-18-10670-t013:** Top-30 features of SVM-RFE ranking for pre-menopausal data partition.

#	Top-30 SVM-RFE (Pre-Menopausal Partition)	
	Feature	Median Rank Position
1	Phenols	10
2	Nulliparous	12
3	Legumes	13
4	METS in 10 years	13
5	QPA	15
6	Magnesium	17
7	Vitamin B1	19
8	Family history of BC	21
9	Fish	21
10	Oil	22
11	Age of menarche	22
12	Contraceptives	24
13	BMI	26
14	Flavonoids	26
15	Total proteins	27
16	Retinoids	27
17	NSAIDs	28
18	RS-141143854	30
19	RS-34402166	34
20	RS-201340741	36
21	RS-71310379	38
22	Phosphorus	38
23	RS-137902538	39
24	RS-141363120	39
25	RS-141420305	39
26	RS-3730581	40
27	RS-200147790	43
28	RS-201652303	43
29	RS-142068825	44
30	RS-145760222	44

**Table 14 ijerph-18-10670-t014:** AUC for different classifiers with the postmenopausal data partition (without performing feature selection).

LR	SVM	MLP	k-NN	BT
0.575	0.594	0.563	0.632	0.570
(±0.065)	(±0.069)	(±0.077)	(±0.068)	(±0.059)

**Table 15 ijerph-18-10670-t015:** Top-30 features of SVM-RFE ranking for postmenopausal data partition.

#	Top-30 SVM-RFE (Postmenopausal Partition)	
	Feature	Median Rank Position
1	Phenols	2
2	BMI	12
3	QPA	14
4	Vitamin B2	15
5	Cereals	17
6	Family history of BC	18
7	Vitamin C	18
8	Vegetables	18
9	Vitamin B1	19
10	Vitamin E	19
11	Oil	20
12	HRT	22
13	Lignans	23
14	Lactation months	24
15	Socioeconomic level	24
16	Vitamin B12	24
17	Vitamin A	25
18	Zinc	25
19	Vitamin D	27
20	Age of menarche	28
21	RS-139697494	28
22	Cholesterol	31
23	Animal protein	31
24	RS-2287499	33
25	Total energy	36
26	RS-5746105	38
27	Stilbenes	39
28	RS-148728256	40
29	RS-201686188	41
30	RS-201498076	42

## Data Availability

Not applicable.

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
