# Peer review of "Evaluation of Feature Selection Techniques for Breast Cancer Risk Prediction"

_ijerph, 2021, doi:10.3390/ijerph182010670_

Round 1
Reviewer 1 Report
This work compares multiple feature ranking strategies with machine learning-based classifiers to find significant characteristics that influence the likelihood of acquiring breast cancer and to improve the performance of breast cancer risk prediction models. The MCC-Spain study provided the dataset, which included 919 cases and 946 controls. Breast cancer is a major public health issue in the United States. The goal of this study is to determine which elements in the cancer risk prediction model are most important in predicting breast cancer. Similarly, before attempting to acquire insight into the data, it is necessary to measure the stability of feature selection algorithms. The performance of multiple feature selection techniques for a set of predictive models is evaluated in this research. Furthermore, their robustness is measured in order to examine the similarity of the feature selection rankings as well as their own stability. In terms of the area under the ROC curve (AUC) metric, the SVM-RFE technique provided the top rating.
Major Issues:
Authors are advised to address the following major issue before resubmission:
- The literature review must include recently published works. In addition, you write the research gap in the existing paper.
- Highlight your contribution using bullet form.
- In this paper, the dataset seems to be unbalanced what is the method you have considered to prevent overfitting and improve model performance and also provide faster and more cost-effective predictive models.
- What is the rationale behind considering the factors considered for feature ranking? Can it be applied to any dataset?
- I can not relate the mathematical formulation with the result. Can you please link them?
- Add explainability in the proposed hybrid model to improve the acceptability.
- Once the feature selection rankings are calculated and iterative performance is measured for each number of features. Show the ranking of the selected features. Explain all the tables carefully.
- The clinical feature plays an important role in the diagnosis of cancer. What is the effect of it on the model?
Reviewer 2 Report
This work approaches the relevant problem of breast cancer risk prediction. The paper is in general well written, but there are some grammatical or typing errors that should be revised by the authors. In the introduction section, for example, there are some sentences or arguments that would need some scientific revision as well, as follows:
- lines 41-42 seem confusing. The condition of sampling to determine whether the tumor is malignant or benign is necessary for any supervised data mining method, such as the ones implemented by the authors in this work;
- lines 48-53 describe recent results on breast images classification with very competitive accuracy, but no clear justification of not using such framework at the current work. The understanding, as described in these lines, is that the problem has been relatively resolved nowadays with CNNs. If not, what are the main drawbacks of such CNN approach? In other words, some additional comments on these results would be necessary to better contextualise the main and novel contribution of this work.
In fact, the novelty is essentially my main concern about this work. The authors have showed results, using standard techniques and analyses and a non-public dataset, with low AUC values (~0,62), which represent performance close to a random classifier, and minor classification improvement (less than 10%) over the use of the full dataset features.
Reviewer 3 Report
This study uses some existing features ranking techniques for breast cancer risk prediction model construction. and validated in a private dataset. I recommend to improve the study with the following revision.
- Please do some statistical significant test for comparison of different classifier performance in terms of AUC values.
- Standard evaluation metric- ROC curve is recommend for evaluation of the result.
- Ref. 4, list of author names need to shorten using et al.,
- use superimpose plot to reduce the paper length, i.e. fig 7
- How is the performance of using F-score measure, https://en.wikipedia.org/wiki/F-score , and PCA for feature selection before applying ML classifier, see detail https://www.techscience.com/iasc/v25n1/39636.
Round 2
Reviewer 1 Report
Authors are advised to incorporate the following points:
In this paper, the dataset seems to be unbalanced what is the method you have considered to prevent overfitting and improve model performance and also provide faster and more cost-effective predictive models.
What is the rationale behind considering the factors considered for feature ranking? Can it be applied to any dataset?
Add explainability in the proposed hybrid model to improve the acceptability.
Once the feature selection rankings are calculated and iterative performance is measured for each number of features. Show the ranking of the selected features. Explain all the tables carefully.
Reviewer 3 Report
- Author should cite the ref papers on "reduction techniques such as PCA or LDA before ML training" (on Breast Cancer or other applications) in the introduction section.
- The title of the paper should revise to reflect the "environmental/ genetic factors",
